# Neuraminidases—Key Players in the Inflammatory Response after Pathophysiological Cardiac Stress and Potential New Therapeutic Targets in Cardiac Disease

**DOI:** 10.3390/biology11081229

**Published:** 2022-08-17

**Authors:** Maren Heimerl, Thomas Gausepohl, Julia H. Mueller, Melanie Ricke-Hoch

**Affiliations:** Department of Cardiology and Angiology, Hannover Medical School, 30625 Hannover, Germany

**Keywords:** neuraminidase, inflammation, cardiomyopathy, heart failure, neuraminidase inhibitors

## Abstract

**Simple Summary:**

In this manuscript, we review research performed on enzymes called neuraminidases (NEUs) and the consequences of their activity on the heart muscle and on a few of its pathophysiological diseases. NEUs are able to cleave off sugars termed sialic acids, which are terminally attached to glycolipids and -proteins. Due to their outermost location, the level of sialic acids affects communication between cells, which in turn affects inflammatory responses. Thus, NEUs hold regulatory features influencing multiple processes within the body. The involvement of NEUs in cardiovascular pathologies i.e., atherosclerosis, myocardial infarction (MI), cardiomyopathies (CMs) and coronary artery disease (CAD) has been investigated and the activity of NEUs or rather the resulting sialic acid level has been identified as a diagnostic biomarker for cardiovascular disease. The downregulation of NEU activity in different animal models diminished inflammation, ameliorated cardiac function and improved vascular health, altogether identifying targeting NEU as a new promising treatment option for cardiac diseases. Fortunately, antiviral drugs blocking the activity of NEUs are already an established therapeutic regime in the treatment of influenza. First clinical trials using NEU inhibitor oseltamivir in patients with chronic heart failure are already ongoing.

**Abstract:**

Glycoproteins and glycolipids on the cell surfaces of vertebrates and higher invertebrates contain α-keto acid sugars called sialic acids, terminally attached to their glycan structures. The actual level of sialylation, regulated through enzymatic removal of the latter ones by NEU enzymes, highly affects protein-protein, cell-matrix and cell-cell interactions. Thus, their regulatory features affect a large number of different cell types, including those of the immune system. Research regarding NEUs within heart and vessels provides new insights of their involvement in the development of cardiovascular pathologies and identifies mechanisms on how inhibiting NEU enzymes can have a beneficial effect on cardiac remodelling and on a number of different cardiac diseases including CMs and atherosclerosis. In this regard, a multitude of clinical studies demonstrated the potential of N-acetylneuraminic acid (Neu5Ac) to serve as a biomarker following cardiac diseases. Anti-influenza drugs i.e., zanamivir and oseltamivir are viral NEU inhibitors, thus, they block the enzymatic activity of NEUs. When considering the improvement in cardiac function in several different cardiac disease animal models, which results from NEU reduction, the inhibition of NEU enzymes provides a new potential therapeutic treatment strategy to treat cardiac inflammatory pathologies, and thus, administrate cardioprotection.

## 1. Introduction

Vertebrate cell surfaces contain a complex array of sugar chains that are bound to lipids and proteins. Sialic acids are a diverse class of α-keto acid sugars characterized by a 9-C-backbone, widely found in animal tissues of all vertebrates [1,2]. They are essential components, positioned at the end of the sugar chains of many glycoproteins, glycolipids and gangliosides at the cell surface or at soluble proteins (Figure 1). Their ubiquitous distribution in glycoconjugates of various origins indicates that a variety of biological functions are associated with them [2]. Sialic acids play important roles in many physiological and pathological processes, e.g., cellular communication, cell assembly and development, mediation of viral and bacterial infections. In particular, they play a crucial role in the development of cardiovascular diseases, as they serve as ligands for leukocytes in the endothelium and are therefore involved in inflammation, atherosclerosis and reperfusion injuries [3,4,5]. NEUs are enzymes able to remove sialic acids, a process which is called desialylation, thus regulating the function of numerous different molecules [6,7].

The endothelial surface layer (ESL) is an important part of the vascular barrier. Coating the intimal surface in blood vessels, thereby creating a barrier separating endothelial cells, blood and neighbouring cells from each other, the ESL plays an important role in various immune reactions, including cardiovascular ones. As one component of the ESL, endothelial cells synthesize the polysaccharide-rich glycocalyx, which surrounds all eukaryotic cells. Consisting of negatively charged sialic acids (monosaccharides), which are in turn bound to proteoglycans, glycolipids and endothelial cells, the glycocalyx converts mechanical into biomechanical signals, thus greatly affecting inter- and intracellular communication and vascular homeostasis. This barrier is also of importance in matters of material and substrate exchange, thereby affecting inflammatory processes as well as vascular permeability, coagulation, blood flow and the complement system [8,9,10,11,12,13]. Individual components of the glycocalyx e.g., heparan sulphate proteoglycans, sialic acids and hyaluronic acid glycosaminoglycans are able to respond immediately to sensed shear stress, which is induced by the blood stream and affects the vessel’s wall, by the physiological production of nitric oxide (NO). NO, known for its vasodilatory effect, consequently modulates the vascular tone [13,14].

In this review, we summarize the present level of knowledge on the importance and role of NEUs and the subsequent level of sialylation in the heart and in the context of the cardiovascular system and certain of its pathologies, i.e., atherosclerosis, myocardial infarction (MI), coronary artery disease (CAD). Further, we classify these findings in terms of possible therapeutic approaches.

## 2. Mammalian Neuraminidases

The polysaccharide-rich glycocalyx can be altered in terms of sialylation by enzymes which are able to recognize sialic acid glycosidic bonds and thus enzymatically remove sialic acids. In mammals, four endogenous NEU enzymes, belonging to the glycoside hydrolase family and also known as sialidases, have been identified to date, namely NEU1, NEU2, NEU3 and NEU4. The catalysed removal of these negatively charged 9-C-backbone sugars, which are terminally attached to glycolipids and glycoproteins on the cell surface, highly affects the cells’ biophysical properties concerning protein-protein, cell-matrix and cell-cell interactions [1,6,15,16,17,18]. Most of the existing literature deals with desialylation by NEUs on the cell surface. However, due to localisation differences of the four NEU enzymes (Table 1), intracellular desialylation also takes place [19,20,21]. All four enzymes are encoded by different genes, own different enzymatic properties and prefer slightly different substrates. In addition, they can be distinguished by their different intracellular localisation [22].

## 3. Functions and Implications of Neuraminidases

NEU1, initially described as a lysosomal protein playing a role in the catabolism of glycosylated proteins [7], is required to form a multienzyme complex with β-Galactosidase (β-GAL) and protective-protein/cathepsin (PPC) A in order to be catalytically active and stable. NEU1 can also homodimerize via the binding site which is usually utilized for binding to PPCA, however, once PPCA is available, the affinity of NEU1 to bind to it instead of binding to another NEU1 is stronger [46]. Further, this conjunction not only protects NEU1 from lysosomal degradation but is required for its transport towards the plasma membrane upon e.g., inflammatory cell activation or differentiation but also for its activation [27,47,48,49]. The gene coding for human NEU1 is located on chromosome 6p21.3 while the single gene for murine NEU1 is located on chromosome 17. Both protein products are 83% identical [23,25,50,51,52]. A reduction in NEU1 usually leads to severe diseases such as sialidosis [53] whereas increased levels stimulate inflammation and phagocytosis [32,54]. Further, NEU1 is a negative regulator of exocytosis [36,37].

There is only little known about the soluble protein NEU2 which is located in the cytosol and able to hydrolyse different glycoproteins, oligosaccharides and gangliosides [24,35]. The gene coding for human NEU2 is located on chromosome 2q37 [24] while its murine counterpart is located on chromosome 1 [26]. The exact role of NEU2 does not seem to be discovered yet. One reason might be lacking evidence which clearly demonstrates glycosidically bound sialic acids in the cytosol or on the inside of the plasma membrane [55], albeit, one study reports complex-type free sialylated N-glycans in the cytosol that were in turn degraded, likely due to the interaction of NEU2 and the cytosolic β-glycosidase (GBA) 3. NEU2 was stabilised by the latter one and both seem to be involved in a non-lysosomal catabolic degradation process [56]. In the physiological context, NEU2 seems to play a role in myoblast differentiation as an increased expression was observed during the differentiation of murine myoblasts. In this regard, experiments using NEU2 overexpressing clones which spontaneously underwent myoblast differentiation, showed that an upregulation of NEU2 per se sufficiently induces differentiation of myoblasts [38,39]. In addition to its role in myoblast differentiation, NEU2 also seems to be involved in the differentiation of neuronal cells derived from a tumour located in the adrenal medulla [40].

NEU3, per se associated with the plasma membrane, has additionally been reported to be translocated towards the plasma membrane in response to different stimuli, e.g., activation and differentiation of immune cells [29,57,58]. The gene coding for human NEU3 is located on chromosome 11q13.5 while its murine counterpart is located on chromosome 7 [24]. Like NEU2, NEU3 is also assigned a role in neuronal differentiation. However, for NEU3, a decreased expression and activity was shown to negatively interfere with cell proliferation while on the other hand promoting neurite extension, thereby inducing a shift towards cells with a differentiated phenotype [41]. In addition, NEU3 was shown to affect cell invasion and focal adhesion of glioblastoma cells, the former one though the regulation of calcium-dependent calpain [42]. Besides, the activation of the enzyme was shown to protect cultured skeletal muscle cells from hypoxia-induced apoptosis through the induction of the EGF receptor signalling pathway and thus, hypoxia inducible factor (HIF) 1α [43].

The gene encoding for human NEU4 is located on chromosome 2q37.3 and its murine counterpart on chromosome 10 [24]. The gene encodes for two different isoforms which holds true for both organisms. These isoforms are NEU4a and b in mice and NEU4L and S in humans, they differ in their first 12 N-terminal amino acids. Murine NEU4a and b have been reported to localize to endoplasmic reticulum (ER) membranes which are calnexin positive. NEU4L and S differ in their subcellular localization with NEU4L being present in mitochondria and lysosomes and NEU4S being present in membranes of the ER. The isoforms differ not only with regard to their localization but also in terms of their sialidase activity with NEU4b displaying a higher activity compared with the shorter isoform NEU4a [24,30,31,58,59]. Under certain conditions, a translocation of NEU4 towards the surface of cells has also been observed, however, the mechanisms behind that movement are not yet understood [24]. It has been proposed that NEU4 is, in connection with NEU3, involved in neural differentiation [60]. NEU4L, expressed in mitochondria as mentioned earlier, seems to be involved in the mitochondrial apoptotic pathway in neuronal cells [44,61].

Human NEU1 bears only a minor resemblance of approximately 19–24% to the other three NEUs with regard to its DNA sequence whereas these exhibit a similarity of 34–40% to each other. In line with this, the binding pocket of NEU1 is also distinct from the one of the other NEU enzymes [24]. However, for an optimal activity, all four human NEU enzymes need an ideal pH environment which is defined between acidic 3.5 and 5.5 [18].

In summary, the four endogenous mammalian NEU enzymes, each encoded by a different gene, exhibit different enzymatic characteristics which subsequently result in partly different substrate specificities as well as different intracellular localization.

## 4. Occurrence and Substrate Specificity of Neuraminidases

All 4 NEUs differ with regard to their expression pattern. NEU1 is ubiquitously expressed and most abundant in kidneys, skeletal muscle, lung, placenta, brain, pancreas, inflammatory cells and cardiomyocytes whereas NEU2 depicts a muscle-specific isoform. NEU3 on the other hand is mostly expressed in the adrenal glands, heart, thymus, skeletal muscle and testis, and NEU4 in the brain, heart, placenta, liver and skeletal muscle [7,27,32,33,52,62]. In mammals, among all 4 NEU enzymes, NEU1 is the most abundant one [15].

The enzymatic removal of sialic acids represents a regulatory function of NEU enzymes through which they highly affect intra- and intercellular communication. The entirety of sialic acid types and underlying connections is versatile due to the fact that these α-keto acids occur in different forms and with different underlying linkages [6,63,64]. Due to these varieties, it is no surprise that the NEU enzymes differ from each other in terms of substrate preferences. Even though NEU1, 2 and 4 share oligosaccharides as substrates, NEU1 prefers those joined with an α2,3 linkage. Sialic acid containing glycosphingolipids, so called gangliosides, belong to the substrate group of all NEU enzymes except for NEU1. Eventually, gangliosides seem to be the only substrates of NEU3, most preferable those with an α2,3 and α2,6 binding. NEU2 and NEU4 share the feature of a broad substrate specificity including, next to already mentioned oligosaccharides, glycoproteins and gangliosides [24,35,58].

## 5. Neuraminidases and Their Role in the Immune System

Sialic acids are able to communicate with other cells and to defend cells against pathogens. One advantage to accomplish these functions is their outermost location on the cell surface. By masking monosaccharides, these are no longer recognised by receptors. The other way around, sialic acids can also mask receptors, can be recognised by receptors or serve as binding sites for lectins and other sialic acid-binding molecules such as hormones or antibodies [2,17,65]. In addition, sialic acids are also utilized by pathogens and tumour cells for immune evasion and their binding is able to promote virulence of bacterial pathogens [6,66,67]. As NEUs can desialylate the cell surface of i.a. active immune cells, they thus own an immunomodulatory feature [1]. In this context, it has recently been shown that the level of sialylation, regulated by NEU enzymes, can directly affect the interaction of natural killer cells with their potential targets [5].

Most of the existing literature about sialidases affecting the immune system discuss the role of NEU1 and only a minor number examines the effect of the other NEUs, a fact underlining NEU1′s function in immune defence. Thus, this part of the review relates primarily to NEU1 and only to a minor extend to the other NEU enzymes.

An increased expression of NEU1 in vitro leads to a pro-inflammatory phenotype of monocytes and macrophages concomitant with an increased release of pro-inflammatory cytokines and increased phagocytosis, caused by a positive feedback-loop with interleukin (IL)-1β (Figure 2). Upon the activation of various immune cells such as macrophages or T-cells or the differentiation of endothelial cells, NEU1 is translocated towards the plasma membrane and is able to modulate the immune system, e.g., by modification of different receptors and adhesion molecules by desialylation [28,32,68,69,70,71,72]. Mice with a NEU1-deficiency showed higher sialylation levels, which was concomitant with less phagocytosis in comparison to non-deficient mice [73]. In line with this, Wang and colleagues confirmed these observations by showing a contrariwise regulation of pro-inflammatory-related M1-like and anti-inflammatory-related M2-like genes with M1-like down- and M2-like upregulated in NEU1-knockout (KO) macrophages isolated from aortic tissue. A reduction of cell surface sialylation to a normal level via exogenous NEU1 restored phagocytosis. NEU1 desialylates cell surface receptors such as Fc receptors for immunoglobulin G, and thus activates phagocytosis in macrophages and dendritic cells [73,74].

Besides, NEU1 is involved in the regulation of toll-like receptors (TLRs), thereby playing a key role in activating the immune response. Membrane-bound TLR4 is highly sialylated on the surface of immune cells. In case of a strong sialylation, Siglec-E can bind to the molecule sending inhibitory signals, which in turn decrease TLR4-signalling. After binding of a ligand such as lipopolysaccharide (LPS) to TLR4, the signalling of G protein-coupled receptors and matrix metalloproteinase (MMP) 9 is increased, subsequently inducing NEU1. NEU1 then forms a complex with MMP9 which binds to TLR4 on the cell surface. The subsequent desialylation of TLR4 prevents binding of Siglec-E and the TLR-pathway is activated given that receptor dimerization can take place. Different response pathways are initiated through e.g., myeloid differentiation primary response protein 88 (MyD88) or tumour necrosis factor α (TNFα). Along that line, an increased TLR4-MyD88 association was detectable after infection and the Mitogen-activated protein kinase (MAPK) pathway was activated. Nuclear factor κB (NFκB) was translocated to the nucleus and nuclear factor NFκB p65 subunit was freely accessible, leading to increased T helper type (Th) 1-cytokine release of IL-12 and interferon-γ (IFN-γ) or TNFα, IL-1β and IL-6 [69,72,75,76,77,78].

Another receptor modified by NEU1 is cell adhesion receptor cluster of differentiation molecule (CD) 44, which is masked by sialic acids to hyaluronic acid and thus is negatively affected in monocytes and T-helper cells. CD44 is involved in cell-cell and cell-matrix interactions and is important for the lymphocyte infiltration into inflamed tissue. SM/J mice characterized by a moderate reduction in NEU1 expression and activity, exhibit a reduced CD44 receptor activity in Th2 cells after the induction of asthma [79,80].

Moreover, in the respective NEU-KO mouse models, NEU1 and NEU3 were identified as pro-inflammatory regulators of leukocyte recruitment upon LPS-stimulation with NEU1 regulating levels of pro-inflammatory cytokines whereas NEU4 exhibits an anti-inflammatory effect, negatively regulating monocyte, neutrophil and natural killer cell infiltration [81]. In human lymphocyte Jurkat T cells, NEU1, NEU3 and NEU4 were determined to be positive regulators of transmigration using cytokines as chemoattractants [82]. In line with this, NEU1 and NEU3 can be found in activated human T-lymphocytes, where they lead to an increased secretion of the pro-inflammatory cytokine IFN-γ and they are able to regulate LPS-induced cytokine production in leukocytes [83,84,85].

Furthermore, NEU1 is linked to autoimmune diseases caused among other things by impaired efferocytosis. Desialylation via NEU1 initiates efferocytosis, thus, in the absence of NEU1, apoptotic cells/debris are not removed anymore [86,87]. Recently, Wu and colleagues revealed a role for NEU enzymes in the influenza A virus-specific CD8^+^ T-cell response by altering the release of infected cells in mice [88].

Only very little literature regarding its role in the immune system can be found related to NEU2. Lee and colleagues propose that NEU2 and NEU3, expressed in cancer cells, play a role the impaired function of natural killer cells, thus helping cancer cells in terms of immune escape [89].

In summary, the literature available demonstrates the importance of NEU enzymes, especially NEU1, their regulatory role and consequences of the latter one in the context of various immune reactions. The role of neuraminidases in this context should not be underestimated since their regulatory features affect a large number of different cell types.

## 6. Neuraminidases in the Context of Cardiac Pathologies

When examining the role of neuraminidases in a certain context, one must automatically imply sialic acids, or rather the level of sialylation versus desialylation, into all considerations. Investigations regarding NEU enzymes and their potential role in the heart and in the cardiovascular context have already been performed. In detail, recent research on the function of NEUs within the cardiovascular system revealed their major involvement and contribution towards different pathologies. This section shall provide an overview on the most impactful mechanisms regarding heart and vessels.

### 6.1. Inflammatory Vascular Diseases: Atherosclerosis, Coronary Artery Disease and Ischemia/Reperfusion Injury

Among all cardiovascular diseases linked to NEUs, atherosclerosis is the best studied one. Atherosclerosis is characterized by fatty degeneration and chronic inflammation at the intima vessel wall, leading to reduced arterial perfusion of downstream organs [90]. When taking place in the vessels of the heart, the permanently impaired energy and oxygen supply of the myocardium results in CAD or worse: the atherosclerotic plaque ruptures and causes MI [91].

The mechanisms mediated by the glycocalyx in combination with the sialic acid metabolism play a role in the pathogenesis of atherosclerosis and vessel disease and provide hope for new therapies, a subject area that has recently been reviewed [12,92]. The most promising targets in the field of sialic acid metabolism are NEUs and this chapter aims to summarize their most important mechanisms from endothelial dysfunction to fatty streak formation to MI. Since NEU1 is, to date, the best studied NEU in this context, NEU1 dependent mechanisms dominate the following chapter.

#### 6.1.1. Atherosclerosis and Coronary Artery Disease

NEUs are involved in the pathogenesis of atherosclerosis *ab initio*: Desialylation of porcine femoral arteries via NEU treatment decreased atheroprotective NO production and impaired reactive vasodilatation during ex vivo exposition to wall shear stress (WSS) [93]. In this context, an in vitro model of endothelial cells exposed to physiological unidirectional shear stress revealed an underlying mechanism: Sia cleavage of these endothelial cells reduced atheroprotective nuclear factor erythroid 2-related factor 2 (Nrf2) signaling including a decrease in phosphorylation of endothelial nitric oxid synthase (eNOS) [94]. Furthermore, microvascular permeability, another contributor towards atherosclerotic endothelial activation [90], was also enhanced following NEU treatment in an in vivo rat model [95]. During atherogenesis, the endothelial dysfunction is followed by the fatal infiltration and oxidation of low density lipoprotein (LDL), a process which is accelerated by elevated LDL plasma levels [90]. Hypomorphic NEU1 expression reduced serum very low density lipoprotein (VLDL) and LDL levels via enhanced LDL uptake of hypersialylated hepatic LDL receptors and decreased hepatic VLDL-triglyceride production in mice [96]. In line with these findings, the VLDL and LDL serum levels of Apolipoprotein E deficient mice (ApoE-/-), a dyslipidemia derived atherosclerosis model, expressing hypomorphic NEU1 were also diminished and atherogenic plaque size was tempered (>50%). Interestingly, the transplantation of hypomorphic NEU1 bone marrow did also reduce aortic plaque lesion size in ApoE-/-mice without affecting serum lipid levels. Decreased recruitment of leukocytes and reduced production of cytokines IL-4 and IFN-γ in the lesion indicate the impaired inflammatory response as the underlying protective mechanism of reduced NEU1 activity [97].

Monocyte recruitment and migration towards the intima is a crucial step in the pathogenesis of atherosclerosis and accompanied by macrophage NEU1 expression in human atherosclerotic plaques [32]. Atherosclerotic inflammation mediators IL-1β and LPS induced NEU1 expression in monocytes in vitro and NEU1 did vice versa. This positive feedback-loop fuels monocyte activation and aggravates monocyte infiltration into the intima [32]. Treatment of mice with elastin-derived peptides (EP), elastin decomposition products of vascular aging, induced atherosclerosis in mice via EP’s directly binding the elastin receptor complex (ERC). NEU1-dependent pro-inflammatory and reactive oxygen species (ROS) generating phosphoinositide 3-kinase-γ (PI3Kγ)-signaling in bone marrow derived cells was diagnosed as the initiating mechanism [98].

As mentioned above, incorporation of LDL and oxidized (ox)LDL into macrophages is substantial for foam cell conversion forming the necrotic core and further escalating vascular inflammation. Desialylation of LDL has been shown to increase macrophage LDL uptake in atherosclerotic lesions of ApoE-/-mice mediated by asialoglycoprotein receptor 1 [99]. Additionally, a proteomic study revealed that plasma membrane NEU1 of THP1-cells but not NEU3 desialysates CD36 [70], a scavenger receptor that mediates the major part of macrophage oxLDL take-up [100]. Increased incorporation of oxLDL in human macrophages after EP stimulation was successfully diminished through the application of sialidase inhibitor 2-deoxy-2, 3-didehydro-N-acetylneuraminic acid (DANA) [70]. Along with raised oxLDL infiltration, vascular smooth muscle cell (VSMC) proliferation in rabbit carotid arteries caused by arterial collaring was intensified in carotid arteries pretreated with NEU [101]. Except for a few in vitro studies reporting decreased MMP9 expression and reduced platelet-derived growth factor (PDGF) and insulin-like growth factor (IGF)-1 receptor activity in VSMC by desialysation [102,103], there is a lack of research regarding the relation of VSCMs and NEUs. In particular when considering that VSMCs compose the fibrous cap as the barrier between necrotic core and the lumen of the vessel [104], thereby protecting from plaque rupture, which in turn can lead to thrombus formation and MI [91].

Inflammation and the size of the necrotic core are major contributors towards plaque rupture [90] and were reduced in several atherosclerosis models by NEU inhibition as summarized above. Targeting NEU could not only be a promising approach to improve vascular health and to reduce the onset of MI but also to reduce myocardial damage and ameliorate cardiac outcome following MI.

#### 6.1.2. Ischemia/Reperfusion Injuries

Inflammation is one of the main elements determining the remodeling process after an ischemic insult, and thus, cardiac outcome. Since monocytes/macrophages feature the wound healing process as well as inflammatory processes, a balance between both is essential [105,106,107,108]. As described above, NEU1 promotes the pro-inflammatory phenotype of monocytes/macrophages [32,74] (Figure 2), thereby affecting immune reactions and myocardial damage following an ischemic insult. It was recently shown that NEU1, upregulated early in the ischemic [27] and infarcted [109] area, significantly contributes to the development of heart failure following ischemia/reperfusion (I/R) injury in mice with a cardiomyocyte-specific overexpression of NEU1. Even though these mice did not show any effects on inflammation, their heart function was declined after I/R injury. Mice with a systemic reduction of NEU1 expression and activity on the other hand displayed better cardiac function and less inflammation after I/R, due to reduced NEU1 expression locally in the heart and in invading immune cells [27]. The positive impact of a reduced NEU1 expression was additionally shown in mice with a cardiomyocyte-specific KO of NEU1, showing reduced cardiac remodeling and oxidative stress following MI. Further, NEU1 deficient mice also revealed an enhanced survival (Figure 3) [109].

A region-specific inhibition of NEU1 restored cardiac function by amelioration of the mitochondrial energy metabolism and reduction of mitochondrial oxidative stress at once, both by the regulation of sirtuin-1/peroxisome proliferator-activated receptor γ coactivator α (SIRT1/PGC-1α) [109]. In line with these results, an increased NEU activity has been shown years ago in the plasma of patients suffering from MI [110,111] as well as increased NEU1 gene expression levels in peripheral blood mononuclear cells (PBMCs) that were isolated from MI patients [32]. In addition, Zhang and colleagues provided data showing that Neu5Ac levels are distinctly increased in the plasma of patients suffering from CAD. By binding to Ras homolog family member (Rho) A and Cell division cycle (CDC) 42, thereby promoting a protein kinase signaling pathway, Neu5Ac triggers myocardial injury, which was shown in vitro in neonatal rat ventricular myocytes as well as in vivo in rats. In line with this, lentiviral suppression of NEU1, naturally responsible for the generation of Neu5Ac, reduced ischemic myocardial injury [112].

In summary, NEU enzymes seem to play a crucial role in the development and progression of different cardiovascular diseases e.g., atherosclerosis, CAD and ischemic cardiomyopathy. Due to the improvement which results from NEU downregulation, the inhibition of NEU enzymes, especially of NEU1, depicts new promising treatment options for cardiac diseases, a field of possibilities that will be further discussed below.

### 6.2. Cardiomyopathy and Heart Failure

Despite the vascular and ischemia related pathomechanisms involving NEUs, desialylation processes are also reported to be involved in several features of CM and cardiac remodeling (CR) [113].

The leading mechanism of the heart facing increased workload of cardiac disease is cardiomyocyte hypertrophy, which disburdens the heart in the short-term and results in impaired contractility, excessive oxygen consumption and heart failure development in the long-term [114]. Chen et al. recently discovered augmented cardiac NEU1 levels in patients suffering from hypertrophic CM as well as in hypertrophy models using mice and rats. Whereas cardiomyocyte-specific deletion of NEU1 mitigated the hypertrophic effects of transverse aortic constriction (TAC), NEU1 overexpression intensified the outcome most likely by induction of transcription factor GATA binding protein (GATA) 4 [33].

Another important characteristic of CR is the development of fibrosis as a response to cardiac tissue injury, characterized by elevated connective tissue production [115]. While on the one hand, cardiac fibrosis can protect the heart against rupture after an injury, on the other hand, fibrosis can lead to heart stiffness, which is associated with a reduced cardiac contractility and function. One of the key signaling pathways in the development of fibrosis is the transforming growth factor beta (TGF-ß) pathway [116]. The activation of TGF-ß receptor 1 (R1) is mediated by sphingolipids like ganglioside monosialodihexosylganglioside (GM) 3 [117]. The upregulation of NEU3 in primary human cardiac fibroblasts induced an increased NEU3-dependent desialylation of ganglioside GM3, which led to a diminished TGF-ß R1 activation and collagen I deposition [118].

NEU1 has lately also been recognized as a potential target for the treatment of two forms of CM: diabetic CM (DCM) and doxorubicin (DOX) induced CM (DIC), both named by their trigger [119,120]. In a DCM mouse model induced by streptozotocin (STZ), cardiac NEU1 expression was increased and associated with the induction of cardiac inflammation, apoptosis and fibrosis by inhibition of 5’ AMP-activated protein kinase alpha (AMPKα) phosphorylation via liver kinase b 1 (LKB1), resulting in reduced SIRT3 and superoxid dismutase (SOD) 2 levels. Accordingly, treatment of cardiomyocytes with high glucose induced an upregulation of NEU1 expression. In addition, adeno-associated virus (AAV) 9-shNEU1 mediated myocardial downregulation of NEU1 in the STZ-DCM model protecting the cardiomyocytes from diabetes induced damage and rescued the DCM phenotype [121]. DOX, trigger of DIC, is a chemotherapeutic agent whose strong antineoplastic efficacy is clouded by the onset of CM in the long-term in several experimental and patient studies [120]. Cardiac dysfunction in DOX treated rats was attended by NEU1 overexpression and prevented by co-treatment with NEU1 inhibitor oseltamivir. Increased dynamin-related protein 1 (DRP1) induced mitophagy due to NEU1 overexpression in DOX treated hearts was identified as the underlying mechanism [122].

The regulation of NEUs is involved in CR processes and onset of different pathomechanisms leading to the development of cardiomyopathies and heart failure and might be promising new therapeutic targets in the therapy of cardiomyopathies.

## 7. Neuraminidase Activity and the Resulting Sialic Acid Level as a Diagnostic Biomarker for Cardiovascular Diseases and concomitant Inflammation

In many cases, diseases are associated with immune reactions. The meaning of NEUs in terms of those reactions and subsequent inflammatory processes has been discussed in this review in previous sections. Cardiovascular diseases e.g., MI are closely connected to complex inflammatory processes [123], thus, leading to the assumption that alterations in the glycocalyx through cleavage of sialic acids by NEU enzymes might affect the mechanisms of action. Indeed, already in the early nineties, it was shown that the sialic acid concentration in the serum of patients depicts to be a strong prediction for cardiovascular mortality [124]. In the following years, studies dealing with an increased NEU expression and activity in serum or plasma of patients who suffered from MI were published, even positively correlating elevated sialic acid levels to the severity of the disease itself as well as to already established risk factors and biomarkers [4,32,110,111,125,126,127,128,129,130]. In addition, it was recently shown in a cohort of 2324 patients who underwent coronary angiography that serum levels of Neu5Ac, generated by NEU1, serve as a metabolic marker for CAD and its progression [112]. In patients suffering from heart failure, sialic acid levels are also frequently utilized as markers for systemic inflammation [127,131,132,133] albeit the underlying mechanisms leading to elevated sialic acid levels are not yet fully elucidated. In addition, sialoadhesin, a member of the sialic acid binding IgG-like lectin family restricted to macrophages is also utilized as a biomarker for inflammation [134]. One possible mechanism is an increased shedding of sialic acids from the cells’ surface as a result of cell damage in cardiovascular diseases such as MI [111]. The scenario of increased sialic acid removal from the cell surface would be in accordance with an increased NEU expression and activity after an ischemic insult [27,110].

Sialic acids and in particular Neu5Ac act as potential markers for cardiovascular disease diagnosis and progression, and targeting the regulatory NEUs may serve as a new avenue for therapeutic treatments of cardiovascular diseases.

## 8. The Potential of Neuraminidases as Therapeutic Targets

An established treatment regime against the activity of viral NEUs are NEU inhibitors (NAIs), a class of antiviral drugs, which block the enzymatic activity of NEUs. For example, viral NEUs are essential for the reproduction of the influenza virus [135]. Widely used anti-influenza drugs such as zanamivir and oseltamivir [136], are viral NEU inhibitors, which show the potential to interfere as well with human NEU. In this context, it has been demonstrated that influenza infections characterized by a high NEU activity, were associated with elevated risk of adverse cardiovascular outcomes [137,138]. Zanamivir has already been shown to effectively inhibit the neuraminidase activity of human lymphocytes in vivo [84]. As reported above, several clinical studies demonstrated that N-acetylneuraminic acid acts as potential biomarkers after cardiac disease such as CAD [112]. Therefore, it seems likely that anti-influenza drugs have potential novel indications of cardioprotection by directly inhibiting mammalian NEU in cardiac inflammatory pathophysiologies. The use of FDA-proven existing drugs for new indications in the treatment of cardiac diseases offers several advantages such as shortened development timelines and cost savings.

Recently, several experimental animal studies in cardiac stress and injury models demonstrated that the use of NEU inhibitors mediates beneficial effects on CR processes and cardiac outcome (Table 2) [33,112,113]. The treatment with the FDA-proven antiviral NEU inhibitors zanamivir and oseltamivir is associated with reduced cardiac hypertrophy and fibrosis and improved cardiac function in various hypertrophy, ischemia/reperfusion injury and myocardial infarction small animal models [33,112,113]. In these small animal models, NEU1 seems to play a crucial role in the respective pathomechanism. Therefore, NEU1 is a potential target for pathological CR processes and first novel computer-designed drugs for the inhibition of human NEU1 (e.g., C-09) have been designed and tested in experimental animal models [33]. Table 2 provides an overview of the previous performed experimental animal studies targeting NEUs in cardiac diseases.

The promising results from small rodents from the various cardiac pathological stress models need to be validated with regard to their efficacy in clinical trials. In a first study, the effects of intravenous zanamivir on the cardiac conduction of healthy volunteers has been investigated (NCT01353729) (Table 3). In this study intravenous zanamivir had no effects on the QT and rate-corrected QT intervals and did not alter cardiac repolarization [139]. Therefore, the trial investigators concluded that the treatment with intravenous zanamivir does not require an additional monitoring beyond standard care. These findings need to be proven in diseased patients with cardiovascular pathophysiologies though.

A first ongoing clinical trial is initiated to investigate “the effects of neuraminidase inhibitor oseltamivir in patients with chronic heart failure” in a randomized, open-label, blank-controlled study (NCT05008679, Wuhan, China) in addition to standard heart failure therapy (Table 3). The study is still in the recruiting phase and the estimated study completing date is calculated for January 2024.

Further investigations and large clinical trials are needed to prove the efficacy and safety of antiviral and specific NEU inhibitors in the therapy of cardiovascular diseases.

## 9. Conclusions

This review highlights the importance and role of NEU enzymes in the heart and thus, of the resulting sialylation in connection with the cardiovascular system as well as cardiovascular diseases e.g., atherosclerosis, CAD and MI. Increasing numbers of publications dealing with antiviral drugs inhibiting NEU enzyme activity as well as resulting clinical and experimental studies have been discussed and contextualised. In conclusion, inhibition of NEU opens up an entirely new and promising treatment regime for cardiovascular diseases. Since so far, most is known about NEU1, the inhibition thereof was evaluated in multiple settings and showed beneficial effects on crucial CR mechanisms and a variety of CMs. With reference to the potent effects of NEU1 inhibition regarding atherosclerosis, CAD and MI damage, NEU1 inhibition is an exciting therapeutic approach which can help to improve important risk factors (atherosclerosis, CAD), fatal events (MI, MI/IR damage) and CM itself.

## Figures and Tables

**Figure 1 biology-11-01229-f001:**
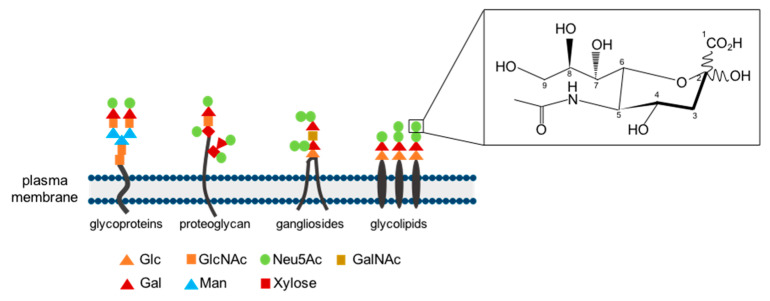
Structure and position of sialic acids: Sialic acids positioned at the end of sugar chains of glycoproteins, proteoglycans and glycolipids. Abbreviations: Gal, galactose; GalNAc, N-acetylglucosamine; Glc, glucose; GlcNAc, N-acetlyglucosamine; Man, mannose; Neu5Ac, N-acetylneuraminic acid. Figure adapted from Klaus et al., 2020.

**Figure 2 biology-11-01229-f002:**
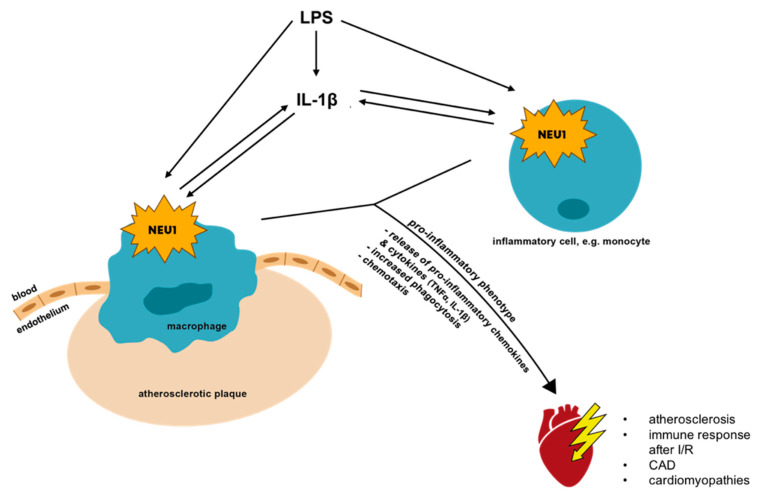
Scheme illustrating the consequences of an altered NEU1 expression in immune cells. NEU1, highly expressed in human macrophages of human atherosclerotic plaques, is upregulated in monocytes and macrophages by LPS and IL-1β, both in turn, inducing the expression of pro-inflammatory cytokine TNFα. Further, there seems to be a positive feedback loop between IL-1β and NEU1, promoting inflammation. The upregulation of NEU1 leads to an enhanced pro-inflammatory phenotype of immune cells concomitant with an increased release of pro-inflammatory chemo- and cytokines, increased phagocytosis and chemotaxis. The pro-inflammatory phenotype contributes to cardiac disease onset and progression. Figure adapted from Sieve et al., 2018. Abbreviations: IL-1β, interleukin-1β; NEU1, neuraminidase 1; LPS, lipopolysaccharide; TNFα, tumour necrosis factor α.

**Figure 3 biology-11-01229-f003:**
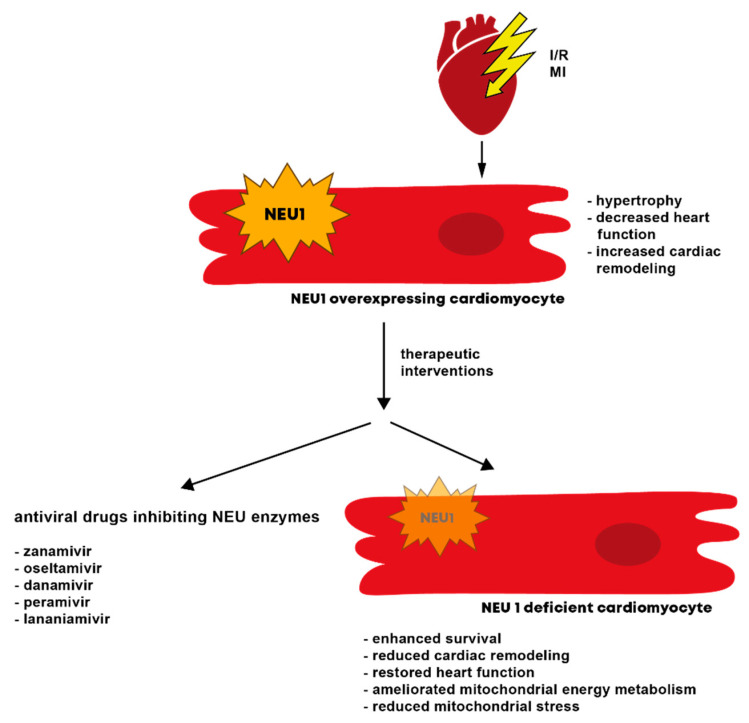
Scheme illustrating the consequences of an altered NEU1 expression on the heart. NEU1 is upregulated in the heart early after I/R and MI. Mice with NEU1 overexpressing cardiomyocytes show hypertrophy, a decrease in heart function and increased cardiac remodelling. Possible therapeutic options to decrease NEU1 expression after an ischemic insult are antiviral drugs inhibiting NEU enzymes or genetic modifications. Mice with NEU1 deficient cardiomyocytes show enhanced survival, reduced cardiac remodelling, a restored heart function, ameliorated mitochondrial energy metabolism and reduced mitochondrial stress. Therefore, the listed antiviral drugs inhibiting NEU enzymes could be a new therapeutic approach with beneficial effects in patients suffering from cardiovascular disease. Abbreviations: NEU1, neuraminidase 1.

**Table 1 biology-11-01229-t001:** Overview of the four different mammalian NEUs and their isoform-specific properties.

Properties of the Four Mammalian NEU Enzymes				
	NEU1	NEU2	NEU3	NEU4
**Human Gene Location**	chromosome 6p21.3 [23]	chromosome 2q37 [24]	chromosome 11q13.5 [24]	chromosome 2q37.3 [24]
**Murine Gene Location**	chromosome 17 [25]	chromosome 1 [26]	chromosome 7 [24]	chromosome 10 [24]
**Subcellular localisation**	Lysosomal, translocation towards the plasma membrane upon different stimuli [7,27,28]	Cytosolic [24]	Associated with the plasma membrane [29]	Murine NEU4a and b, human NEU4S: ER membranes; human NEU4L: mitochondria, lysosomes [30,31]
**Expression pattern**	kidneys, skeletal muscle, lung, placenta, brain, pancreas, inflammatory cells and cardiomyocytes [7,27,32,33]	Muscle-specific isoform [7,34]	Adrenal glands, heart, thymus, skeletal muscle and testis [7]	Brain, heart, placenta, liver and skeletal muscle [31]
**Substrate preferences**	Oligosaccharides with an α2,3 linkage [24]	Oligosaccharides, gangliosides, glycoproteins [24,26,35]	Gangliosides, most preferable with an α2,3 and α2,6 linkage [24]	Oligosaccharides, gangliosides, glycoproteins [24]
**Physiological functions**	Regulates exocytosis, modulator of inflammatory resposnse [27,36,37]	Myoblast and neuronal cell differentiation [38,39,40]	Neuronal cell differentitation, focal adhesion, cell invasion, cell survival, proliferation [41,42,43]	Neural differentiation, mitochondrial neuronal apoptosis [44,45]

**Table 2 biology-11-01229-t002:** Overview of experimental studies targeting NEUs in cardiovascular diseases.

NEU Inhibitors in Experimental Animal Studies			
Drug	Model	Effects	References
Zanamivir (antiviral drug)	(1) Transverse aortic constriction (TAC)-induced cardiac hypertrophy (mice and rat model) (1) Isoproterenol (ISO)-induced hypertrophic rat models (2) Myocardial ischemia rat model (2) ISO-induced injury rat model, permanent left anterior descending coronary artery ligation	(1), (2) beneficial effects after post-treatment, reduced cardiac, hypertrophy, fibrosis, improved cardiac function	(1) [33] (2) [112]
Oseltamivir (antiviral drug)	(1) TAC-induced cardiac hypertrophy (mice and rat model) (1) ISO-induced hypertrophic rat models (2) Myocardial ischemia rat model (2) ISO-induced injury rat model, permanent left anterior descending coronary artery ligation (3) Doxorubicin (DOX) induced cardiomyopathy rat model (4) ISO- and Angiotensin (ANG)II-induced heart failure mouse models	(1), (2) beneficial effects after post-treatment, reduced cardiac, hypertrophy, fibrosis, improved cardiac function (3) beneficial effects after pre- and co-treatment (4) beneficial effects of post-treatment	(1) [33] (2) [112] (3) [122] (4) [113]
Compound-C09 (NEU1 inhibitor)	(5) TAC-induced cardiac hypertrophy	(5) beneficial effects after post-treatment, reduced cardiac, hypertrophy, fibrosis, improved cardiac function	(5) [33]
Danamivir	(6) Apolipoprotein E deficient (Apo E-/-) atherosclerosis mouse model	(6) reduced atherosclerotic lesion size	(6) [99]
C9-BA-DANA	(7) Apo E-/-atherosclerosis mouse model	(7) reduced atherosclerotic lesion size	(7) [99]
CG17701	(8) Apo E-/-atherosclerosis mouse model	(8) reduced atherosclerotic lesion size	(8) [99]
CG14601	(9) Apo E-/-atherosclerosis mouse model (1) (10) Low density lipoprotein receptor (Ldlr)-/-and high fat diet atherosclerosis mouse model	(9), (10) reduced atherosclerotic lesion size	(9), (10) [99]
CG22601	(11) Apo E-/-atherosclerosis mouse model (12) Ldlr-/-and high fat diet atherosclerosis mouse model	(11), (12) reduced atherosclerotic lesion size	(11), (12) [99]

**Table 3 biology-11-01229-t003:** Overview of clinical trials targeting NEUs in cardiovascular diseases.

NEU Inhibitors Targeting Cardiovascular Disease in Clinical Trials			
Drug	Study Description	Effects	Trial Registration
Zanamivir (antiviral drug)	Effect of intravenous zanamivir on cardiac conduction in healthy volunteers (*n* = 40); randomized controlled trial	Completed: Intravenous zanamivir does not affect cardiac repolarization	ClinicTrials.gov NCT01353729
Oseltamivir (antiviral drug)	the effects of neuraminidase inhibitor oseltamivir in patients with chronic heart failure (*n* = 388); randomized, open-label, blank-controlled study	Ongoing	ClinicTrials.gov NCT05008679

## Data Availability

Not applicable.

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
