# Peer review of "Neuraminidases—Key Players in the Inflammatory Response after Pathophysiological Cardiac Stress and Potential New Therapeutic Targets in Cardiac Disease"

_biology, 2022, doi:10.3390/biology11081229_

Round 1

Reviewer 1 Report

The manuscript is well written and adequately cited. I want to congratulate the authors for writing this kind of informative review. this manuscript can be accepted in this format

Reviewer 2 Report

The manuscript summarized the types of neuraminidases (NEUs) and their functions, focusing on the role of NEU1 in the cardiovascular system and cardiac diseases. Based on a comprehensive description of research papers with in vitro, in vivo, and clinical data, the authors proposed that NEUs activity and the resulting sialic acid level can be used as diagnostic biomarkers for cardiovascular diseases, and drugs aimed at inhibiting NEUs are promising as a novel treatment.

The manuscript is thorough and detailed in respect to the completeness of the topic covered. It introduced NEUs and their role in the immune system first, then dived into cardiac pathologies and their potentials as biomarkers and therapeutic targets. When searched through PubMed, there is no similar review published in recent 10 years, which is a strong indication for the novelty of the paper. At the last parts of the paper, the author listed the effects of NEUs inhibitor in small rodents and identified a space for future study in clinical trials. The references are closely related to the topic and 50 out of 171 of them are within 5 years. It is reasonable for some of the reference to be in nineties and early 2000 since the authors were intended to describe the development of the research in the field (e.g. line 400). Below are some suggestions for the improvement:

1. At page 2 (part 3) and page 3 (part 4), when talking about functions and implications of NEUs, and occurrence and substrate specificity, a table will be super helpful for presenting the data. For example, the table can contain NEU type, gene location, protein location, protein expression pattern, substrate preference, function, and reference.

2. Figure 1 is hard to understand. One of the reasons is that the description of Figure 1 is scattered in the main context as it was mentioned for four times at line 177, line 213, line 322 and line 333 respectively. The other reason is that some of the components in the figure were also mentioned multiple times, with or without “Figure 1” marked nearby. For example, LPS was put in the figure to indicate its stimulation effect, described at line 212 and 219. However, it was also mentioned at line 193 for the involvement of NEU1 in TLRs regulation. It is confusing if they are at a same pathway or totally different. Thirdly, some parts of the figure were not mentioned in the main context. For example, there is an arrow pointing from LPS to IL-1ß, which is possibly from the pathway described at line 193, but not indicated in the context. Because of this arrow, it is even more confusing if line 193 and line 212 are the same thing, and if they are, there exists an over-simplified problem of the figure. In addition, it is not clear what is the relationship between NEU1, ß-GAL and PPCA as drawn in the monocyte. Again, there is a short description at line 92, but with no indication. Plus, why macrophage does not contain such complex? The arrows from monocytes to TNFÉ‘, IL-1 ß and NO, then to heart do not have clear explanation, and the arrow bearing pro-inflammation phenotype from macrophage and monocyte to NEU1 overexpressing cardiomyocyte also needs more clarification. There either need to be more description of the figure in the legend, or in the main context when it first appears.

3. At page 8, the hierarchy bullet number of “Cardiomyopathy and Heart Failure” should be 6.2, instead of 6.1.3.

4. Here are two more recent papers that are related to the field:

·       Betteridge, K. B.; Arkill, K. P.; Neal, C. R.; Harper, S. J.; Foster, R. R.; Satchell, S. C.; Bates, D. O.; Salmon, A. H. J. Sialic acids regulate microvessel permeability, revealed by novel in vivo studies of endothelial glycocalyx structure and function. J. Physiol. 2017, 595, 5015– 5035, DOI: 10.1113/JP274167

·       Bourguet, E.; Figurska, S.; Fra̧czek, M.M. Human Neuraminidases: Structures and Stereoselective Inhibitors. J Med Chem. 2022, 65(4), 3002-3025, DOI: 10.1021/acs.jmedchem.1c01612

Reviewer 3 Report

This is a well-organized paper summarizing the function of neuraminidases, especially in cardiac system. This manuscript covers the recent findings about the importance and role of NEUs and the subsequent level of sialylation in the heart and in the context of the cardiovascular system. Therefore, this manuscript should be published at Biology but not before some minor adaptations.

Here are several suggestions which may help.

Q1: In line 12-13, the author described that “NEUs are able to cleave off sugars termed sialic acids, which are terminally attached to glycolipids and proteins on the outside of cells in the body”. Please check whether proteins inside the cell (e.g., lysosome) could also be desialylated by sialidase.

Q2: In part 3 “Functions and Implications of Neuraminidases” and 4 “Occurrence and Substrate Specificity of Neuraminidases”, It would be better if the author summary the function, occurrence and substrate specificity of NEUs in a table or simple figure.

Reviewer 4 Report

This review by Heimerl et al. thoroughly reviewed the roles of NEUs (neuraminidases) in cardiovascular diseases. In general, the manuscript is clearly written and summarized the most recent advances in the field. While reading smoothly, it could be further improved if the authors could address the minor comments here:

1. Please provide a cartoon to show the structures of Salic acids. Also, please show where is the Salic acid on the glycolipids, glycoproteins, and gangliosides. Also a cartoon of the structure of the NEUs showing the binding pocket for Salic acid at the end of the complex.

2. Please provide a cartoon to show how Salic acid could block the function of receptors or signaling molecules and the removal of it could activate the receptor/signaling molecule. And how NEUs play a role in immunomodulation.

3. Please offer insights into the physiological roles of NEUs in the cell.

4. In line 198, is “trough” a typo for “through”?

5. In line 387, what does “Immoderate” mean?
